# Development of Biocompatible Digital Light Processing Resins for Additive Manufacturing Using Visible Light-Induced RAFT Polymerization

**DOI:** 10.3390/polym16040472

**Published:** 2024-02-08

**Authors:** Mauricio A. Sarabia-Vallejos, Scarleth Romero De la Fuente, Pamela Tapia, Nicolás A. Cohn-Inostroza, Manuel Estrada, David Ortiz-Puerta, Juan Rodríguez-Hernández, Carmen M. González-Henríquez

**Affiliations:** 1Facultad de Ingeniería, Arquitectura y Diseño, Universidad San Sebastián, Santiago 8420524, Chile; mauricio.sarabia@uss.cl (M.A.S.-V.); dortiz5@uc.cl (D.O.-P.); 2Departamento de Química, Facultad de Ciencias Naturales, Matemáticas y del Medio Ambiente, Universidad Tecnológica Metropolitana, Santiago 7800003, Chile; scarleth.romerod@utem.cl (S.R.D.l.F.); pamela.tapiah@utem.cl (P.T.); 3Programa Institucional de Fomento a la Investigación, Desarrollo e Innovación, Universidad Tecnológica Metropolitana, Santiago 8940000, Chile; 4Programa de Fisiología y Biofísica, Facultad de Medicina, Universidad de Chile, Santiago 8389100, Chile; nicolas.cohn@utem.cl (N.A.C.-I.); maestrada@uchile.cl (M.E.); 5Polymer Functionalization Group, Departamento de Química Macromolecular Aplicada, Instituto de Ciencia y Tecnología de Polímeros-Consejo Superior de Investigaciones Científicas (ICTP-CSIC), 28006 Madrid, Spain; jrodriguez@ictp.csic.es

**Keywords:** DLP resin, bone scaffold, bioceramics, RAFT polymerization, photoabsorbers

## Abstract

Patients with bone diseases often experience increased bone fragility. When bone injuries exceed the body’s natural healing capacity, they become significant obstacles. The global rise in the aging population and the escalating obesity pandemic are anticipated to lead to a notable increase in acute bone injuries in the coming years. Our research developed a novel DLP resin for 3D printing, utilizing poly(ethylene glycol diacrylate) (PEGDA) and various monomers through the PET-RAFT polymerization method. To enhance the performance of bone scaffolds, triply periodic minimal surfaces (TPMS) were incorporated into the printed structure, promoting porosity and pore interconnectivity without reducing the mechanical resistance of the printed piece. The gyroid TPMS structure was the one that showed the highest mechanical resistance (0.94 ± 0.117 and 1.66 ± 0.240 MPa) for both variants of resin composition. Additionally, bioactive particles were introduced to enhance the material’s biocompatibility, showcasing the potential for incorporating active compounds for specific applications. The inclusion of bioceramic particles produces an increase of 13% in bioactivity signal for osteogenic differentiation (alkaline phosphatase essay) compared to that of control resins. Our findings highlight the substantial improvement in printing precision and resolution achieved by including the photoabsorber, Rose Bengal, in the synthesized resin. This enhancement allows for creating intricately detailed and accurately defined 3D-printed parts. Furthermore, the TPMS gyroid structure significantly enhances the material’s mechanical resistance, while including bioactive compounds significantly boosts the polymeric resin’s biocompatibility and bioactivity (osteogenic differentiation).

## 1. Introduction

The global healthcare scenario is profoundly impacted by the demanding issue of finding alternatives to substitute damaged tissues or organs caused by trauma, diseases, or congenital conditions. Among these challenges, bone injuries that surpass the natural ability to self-heal become a formidable obstacle [1]. Given the global increase in the aging population and the worldwide escalating obesity pandemic, there is projected to be a significant rise in the incidence of acute bone injuries in the upcoming years [2].

In cases where the natural self-recovery mechanism of bone tissue is limited due to its extensive area, artificial alternatives are required to replace the original tissue, particularly in critical situations. To address these challenges, artificial bone-engineered alternatives have been proposed as tissue engineering (TE) solutions [3]. TE was defined in 1993 by R. Langer and J.P. Vacanti as “an interdisciplinary field that aims the development of biological substitutes that can be used to replace, restore or improve tissue function” [3]. There are several criteria that a biomaterial must fulfill to be considered a promising material for bone replacement: (1) it must present high biocompatibility, (2) it must promote cell and tissue adhesion, and (3) it should generate gene induction in the surrounding media [4]. Notably, in the case of bone tissue, the material must act as a guide for connective tissue and bone growth. Therefore, it becomes necessary to present an interconnected network of microchannels or pores that allows vascularization, innervation, and nutrient supply [5]. It also needs to present appropriate mechanical resistance to withstand the loads and strains that bone tissue is usually subjected to in physiological conditions. With the finality to improve the mechanical properties of the material, in addition to its bioactive characteristics, several authors have included bioceramic particles in implants as fillers. The most-used fillers for these purposes are nano-hydroxyapatite (nHA), β-tricalcium phosphate (β-TCP), and bioactive glass (BG). nHA has been used for several years as an artificial substitute for bone scaffolds due to its high biocompatibility and osteoconductivity, owing to its chemical similarity with natural bone [6,7], which makes bone cells adhere to its surface, and, therefore, proliferate and mineralize on the scaffold. On the other hand, among the calcium phosphates, β-TCP is preferred to fabricate bone scaffolds since it can actively induce bone ingrowth and has an adequate degradation rate [8].

One of the TE strategies is imparting specific chemical or physical functionalities to the materials with the finality to fulfill the abovementioned criteria. Another way to improve these characteristics is to develop innovative designs or implement new fabrication methods for cellular scaffolds. Additive manufacturing (AM), or 3D printing, has emerged as an innovative procedure that enables the creation of 3D-printed objects without the need for expensive manufacturing processes. Also, this technology allows the creation of complex parts with internal and external architectures that could not be created using other manufacturing methods. AM reduces costs and empowers individuals or industries to design and produce goods [9]. This aligns with Toffler’s concept of the “prosumer” [10], wherein consumers become producers. As a result, the established firms face increased competition and challenges from the advancements of AM technologies [11].

Since the initial inception of AM, significant endeavors have been devoted to extending its scientific and technological influence within academic and industrial environments [12]. Among the numerous techniques and equipment available for AM, there is an increasing interest in photopolymerization-based methods, such as stereolithography (SLA) and digital light processing (DLP), due to the natural flexibility that they impart to polymer chemistry [13,14]. Due to its ability to tolerate a wide range of monomers and functional groups, radical polymerization has emerged as the predominant mechanism for synthesizing polymer networks. Photopolymerization primarily relies on utilizing non-living free radical polymerization to solidify liquid monomers/oligomers when exposed to a light source with a specific wavelength [15]. Nevertheless, employing conventional free radical polymerization in AM forms inert or “dead” polymers that cannot be further modified [12], offering limited control over chain growth. Also, most of the reactives and photoinitiators used in the conventional radical photopolymerization approach involve partially cytotoxic agents [16,17], thus diminishing its application for the fabrication of biomedical devices. An interesting potential approach to impart a “living character” to crosslinked materials that control polymer architectures is reversible deactivation radical polymerization (RDRP), which enables the reversible deactivation of propagating radical species. One of these techniques is called reversible addition–fragmentation chain-transfer (RAFT) polymerization [18,19], and using RAFT agents, such as trithiocarbonates (TTCs), into the polymer network, makes it possible to reactivate the polymers under UV/visible light exposure, allowing for the incorporation of new monomers into an existing network, thus facilitating the formation of dynamic polymer networks [20,21]. One of the main advantages of RAFT methodology is that most of the reactives used in this synthesis route are biocompatible, thus making it possible to fabricate “bio-friendly” scaffolds. Moreover, when opting for the RAFT method as the primary synthesis approach, another pivotal aspect emerges. This method’s reversibility means that the resulting polymer bonds are comparatively weaker than those typically achieved in traditional polymerizations. Consequently, the resulting materials exhibit enhanced biodegradability, adding another dimension to their functionality.

One of the main problems with RAFT synthesis is their non-tolerance to oxygen-rich environments, making it a problematic polymerization route. Therefore, some obstacles must be addressed to advance the application of RAFT-enabled 3D-printing processes. Firstly, finding a solution where oxygen does not hinder the polymerization process; secondly, achieving rapid kinetics at room temperature becomes essential for the efficient photopolymerization of the resins used for SLS or DLP technologies. Consequently, to create a system that can freely operate in an open-air environment, there is a need for an effective photoRAFT system that can tolerate oxygen [22,23]. Until now, various approaches have been employed to accomplish controlled radical polymerization systems that are tolerant to oxygen. Among these strategies, photoredox catalysts with a high photo-reducing capability, such as eosin Y (EY) [18], have emerged as a solution. EY can transform triplet oxygen into singlet oxygen states, which enables oxygen trapping, thus facilitating oxygen-tolerant systems. This technique is called photoelectron/energy transfer RAFT (PET-RAFT) polymerization.

As mentioned before, one of the main issues or criteria that must be fulfilled for bone scaffolds is to possess a highly interconnected pore network to assure cellular growth while maximizing the mechanical resistance of the 3D-printed pieces, even though the material loss means having a high porosity. This tradeoff between porosity and mechanical resistance needs to be precisely optimized to ensure the outstanding performance of the material. Triply periodic minimal surfaces (TPMSs) have emerged as a potent tool for the design of porous TE scaffolds [24,25] due to their remarkable advantages over conventional scaffolds like their smooth surfaces, highly interconnected structures, and the ability to control porosity precisely [26]. These biomimetic structures can be found in natural architectures such as leaf surfaces and butterfly wings [27]. Compared to the more commonly used lattice scaffolds in similar conditions, TPMS scaffolds demonstrate enhanced permeability and superior mechanical properties, making them highly valuable for TE scaffolds [28].

In this study, our research group aims to design and fabricate cellular scaffolds based on synthetic resins for bone regeneration with improved biocompatibility, bioactivity, and mechanical performance. With the finality to enhance these characteristics, two different methodologies were applied. Firstly, the generation of complex internal channels based on TPMS structures fabricated via DLP 3D-printing technology, and secondly, the inclusion of an inorganic bioceramic phase with the finality of both increasing the material’s bioactivity and improving the printed piece’s mechanical resistance. The synthesized resin was characterized by its printability via its rheological behavior. Then, physico-chemical and mechanical tests were performed on the printed parts with the finality to ensure their performance as possible materials for tissue engineering applications. Finally, biocompatibility and osteogenic differentiation assays were performed on the scaffolds.

## 2. Materials and Methods

### 2.1. Materials

2-hydroxyethyl methacrylate (HEMA, 97%); 2-(Dimethylamino)ethyl methacrylate (DMAEMA, 97%); Acrylamide (AAm, 98%); 2-Hydroxypropyl methacrylate (HPMA, 97%); and poly(ethylene glycol) diacrylate (PEGDA), with an average molecular weight of 250 g mol^−1^; were used as monomers. Additionally, Triethanolamine (TEOH, 99%), Eosin Y disodium salt (EY, 90%), and Rose Bengal (RB, 80%) were utilized as reducing, photoinitiator, and photo-absorbing agents, respectively. All the compounds mentioned above were acquired from Sigma Aldrich (St. Louise, MO, USA). 

4-cyano-4-[(dodecyl sulfanyl thiocarbonyl)sulfanyl] pentanoic acid (CDTPA, 97% HPLC) and S-S dibenzyl trithiocarbonate (DBTTC, 97% HPLC) were utilized as RAFT agents and were purchased from Sigma Aldrich (St. Louise, MO, USA). Ethanol (C_2_H_6_O, 99.9%), Isopropanol (C_3_H_8_O), and Dimethyl Sulfoxide (C_2_H_6_OS) were acquired from Merck (Darmstadt, Germany). HPLC-water for chromatography was obtained from water purification systems (Symplicity, Merck KGaA, Darmstadt, Germany). Finally, Glycerol (C_3_H_8_O_3_) and Diiodomethane (CH_2_I_2_) were purchased from Sigma Aldrich (St. Louise, MO, USA). 

For the nano-hydroxyapatite (nHA) synthesis, ammonium (NH_4_) solution 25%, calcium nitrate tetrahydrate (Ca(NO_3_)_2_·4H_2_O), and ammonium dihydrogen phosphate ((NH_4_)H_2_PO_4_) from Merck (Darmstadt, Germany) were used. Subsequently, these particles were sifted using a mesh N° 230 (63 µm opening, Gilson ASTM-U certified). On the other hand, sodium phosphate dibasic dodecahydrate (Na_2_HPO·12H_2_O) (from Merck KGaA, Darmstadt, Germany) and calcium chloride dihydrate (CaCl_2_·2H_2_O) (from Winkler S.A., Santiago, Chile) were used together with ammonium (NH_4_) solution 25% for the synthesis of β-tricalcium phosphate (β-TCP).

For the biological studies, MC3T3-E1 (ATCC, Manassas, VA, USA) cells and a Dulbecco’s Modified Eagle’s Medium (DMEM) with 2-[4-(2-hydroxyethyl)-1-piperazinyl] ethanesulphonic acid (HEPES), penicillin, streptomycin, amphotericin B (all from Sigma-Aldrich, St. Louise, MO, USA), and fetal bovine serum-FBS- Gibco^TM^, from ThermoFisher (Waltham, MA, USA), were used. Cell viability was determined using the AlamarBlue HS^®^ reagent from ThermoFisher (Waltham, MA, USA), and ALP assay was performed using a kit (ABCAM, ab83369, Cambridge, UK).

### 2.2. Equipment

3D printing was performed in a photon mono 4K, AnyCubic (Shenzhen, China), which has a printing accuracy of 3840 × 2400 px; a layer thickness of 0.01–0.15 mm; a horizontal resolution of 35 µm; and a printing speed of ≤5 cm/h (0.97 in./h). The light source was modified with the finality to change the wavelength to 525 nm (15 LEDs of 3 W, in total 45 W, 3500–4500 µW/cm^2^). The scaffolds were cured through radiation exposure using a homemade lamp of 525 nm (12 LEDs of 3 W, in total 36 W, ~150 mW/cm^2^). 

A rheometer model MCR 72, Anton Paar (Graz, Austria), was used to perform the rheological assessment for all the synthesized resins. The parallel-plate geometry PP 25 (with a diameter of 25 mm) was employed for viscosity measurements using the dynamic oscillatory shear tests. The gap between the plates was set at 250 μm using a controlled temperature of 25 °C with the Peltier-controlled Plate System (CoolPeltier^TM^).

UV/Vis Spectrophotometer model Jasco V-730 (Tokyo, Japan) was used to determine the necessary resin irradiation time and the optimal polymerization wavelength. This instrument has a double beam with a wavelength range from 190 to 1100 nm, a small footprint, 1 nm SBW, and a wide dynamic range.

Water contact angle measurements were performed using a Theta lite optical tensiometer from Attension-Biolin Scientific (Gothenburg, Sweden), adding 4 μL of liquid phase (water, glycerol, or diiodomethane) over the solid sample. 

FT-IR was used to determine the chemical composition of the compounds in an FT/IR-4600 Jasco International Co., Ltd. (Tokyo, Japan), whose standard working range is from 7800 to 350 cm^−1^. Confocal Raman spectrometer/AFM CRM-Alpha 300 RA (WITec GmbH, Ulm, Germany) with an Nd:YAG laser (50 mW to 532 nm) (Ulm, Germany) were utilized to corroborate the signals of the resins.

The mechanical tests were performed in a Zwick–Roell universal test machine, model ProLine Z005 (Ulm, Germany). A 2500 N load cell was mounted on the system, and a 2.5 mm/min testing speed was used during the measurements, which is low enough to be considered in the quasistatic regime.

For biological studies, absorbance was measured at 570 nm for viability and 405 nm for ALP assay using a microplate reader BioTek Synergy HTX from Agilent Technologies Inc. (Santa Clara, CA, USA). Thermo Scientific^TM^ Forma^TM^ Serie 3 Water Jacketed CO_2_ Incubator, 184 L (Thermo Scientific, Ashville, NC, USA), was utilized to maintain cells.

### 2.3. Methodology

#### 2.3.1. Scaffold Design and Printing

For scaffold fabrication and subsequent 3D printing, we employed triply periodic minimal surfaces (TPMSs). TPMSs are smooth and porous surface models with the distinctive property of infinite repetition in three spatial directions with zero mean curvature at every point [29]. The computational generation of TPMS-based scaffolds can be accomplished by considering the sublevel $\Omega ≔\left\{\left(x,y,z\right):\varphi \left(x,y,z;\;\tau ,c\right)\leq 0\right\}$. Here, spatial points within the sublevel set Ω collectively constitute a solid region, where the implicit surface φ delineates its boundary. Also, τ and c constitute the parameters governing the pore size r and the porosity ρ, respectively. See [30,31] for a detailed discussion about TPMS and the porosity obtained. In this study, we employed the following TPMSs as the implicit functions to define the scaffolds [24]:(1)$$\varphi_{G}\left(x,y,z\right)=\sin\left(2\pi \tau x\right)\cos\left(2\pi \tau y\right)+\sin\left(2\pi \tau z\right)\cos\left(2\pi \tau x\right)+\sin\left(2\pi \tau y\right)\cos\left(2\pi \tau z\right)-c$$
(2)$$\varphi_{D}\left(x,y,z\right)=\cos\left(2\pi \tau x\right)\cos\left(2\pi \tau y\right)\cos\left(2\pi \tau z\right)-\sin\left(2\pi \tau x\right)\sin\left(2\pi \tau y\right)\sin\left(2\pi \tau z\right)-c$$
where G and D stand for Gyroid and Diamond TPMSs, respectively, for comparative purposes, and we used the scaffold proposed in [32], consisting of a cylinder of 10 mm of height H and diameter D, with straight pores of size r=1 mm, and a porosity of ρ~40%. To create the TPMS scaffolds, we delimited them to a cylindrical volume of interest with the same height and diameter as the straight pore scaffold. Also, we define the porosity as the linear expression $\rho =m_{ms}c+b_{ms}$, where $m_{ms}$ and $b_{ms}$ are parameters that depend on the TPMS Equations (1) and (2). These parameters are estimated using a linear polynomial fitting from the porosity obtained from TPMS scaffolds created using different values of c. For the pore radii, we used the expression $r=\frac{1}{a\tau }\left(1-c\right)$, where a=8 for the D-TPMS, and a=4 for the G-TPMS. From these expressions, τ and c can be calculated by replacing the desired values of ρ and r and solving the list of equations.

To add a shell to the TPMS scaffolds, we use the hybridization of TPMS scaffolds [29] and define a new implicit surface $\varphi_{h}=\gamma \varphi_{S}+\left(1-\gamma \right)\varphi_{TPMS}$, where $\gamma \left(x,y,z\right)$ is a smooth and spatial transition function from 0 to 1, and $\varphi_{S}$ is the implicit function for the shell. For the function $\gamma \left(x,y,z\right)$, we modified the tapered window from [33] defined by the following:(3)$$\Pi \left(\xi ;c\right)={\;}_{0.5e^{\left(1-\left|\xi \right|/c\right)/0.03}\;\;for\;\left|\xi \right|>c}^{1-0.5e^{-\left(1-\left|\xi \right|/c\right)/0.03}\;for\;\left|\xi \right|<c}$$
where $\left|\cdot \right|$ is the Euclidean norm to obtain a cylindrical shape, i.e.,
(4)$$\gamma \left(x,y,z\right)=\Pi \left(x,y;\;\bar{R}\right)\cdot \Pi \left(z;\;\bar{h}\right)$$
where $\bar{R}=R-t$, $\bar{h}=\frac{h}{2}-t,$ and t is the thickness of the shell. Finally, the implicit function for the shell $\varphi_{S}$ is defined as $\varphi_{S}\left(x,y,z\right)=\max\left\{x^{2}+y^{\;2}-R^{\;2},\;z-\bar{h},\;-z+\bar{h}\right\}$. We remark that in the case of the hybrid TPMS scaffold, the implicit function is centered around the z-axis, i.e., $z\in \left[-\frac{h}{2},\;\frac{h}{2}\right]$. All algorithms were implemented in an in-house code using Python 3.9 and the package Pygalmesh v0.10.7 [34].

#### 2.3.2. Synthesis of the Biocompatible Resins

Two distinct types of resins were synthesized, varying the concentrations of PEGDA_250_ and the monomers. Thus, the first resin is constituted by DMAEMA and AAm as monomers and DBTTC as the RAFT agent (Table 1). We used HEMA and HPMA as base monomers for the second resin and CDTPA as the RAFT agent (Table 2). The crosslinking agent and monomers were purified using a basic alumina column to eliminate the hydroquinone-polymerization inhibitor. Eosin Y was utilized as a photoinitiator, Rose Bengal (RB) as a photoabsorber to increase printing resolution and accuracy (see Appendix A), and TEOH as a reducing agent for both reaction mixtures. 

Table 1 shows the mole ratios of PEGDA_250_, DMAEMA, AAm, DBTTC, EY, and RB used for the PDAD resins. The reaction mixture (resin) was vortexed and sonicated until complete dissolution was achieved. The synthesis was based on the study reported by Bagheri et al. (2020) [35]. 

##### Resin Composed of [PEGDA_250_]:[HEMA]:[HPMA]:[CDTPA]:[EY]:[RB]: [TEOH], Namely, PHHC

Similar to the previous case, the mole ratios of PEGDA_250_, HEMA, HPMA, CDTPA, EY, and RB used to synthesize PDAD resins are shown in Table 2, as reported by Yanan Li et al. [36].

PDAD-8 and PHHC-3 resins (Table 2) were selected because of their low contact angles below 30° and viscosity values below 5 MPa∙s; and these characteristics have been reported to improve the printing process due to their improved wettability between 3D-printed layers and, as a result, the adhesion forces. In addition, these resins presented biocompatibility in the range of the control sample. These resins were used to manufacture a solid cylinder of 10 mm diameter and two different types of cylinders with inner channels (gyroid and diamond; see Appendix A). These channels were created to increase and promote cell adhesion, with the finality of emulating the bone’s natural structure. 

After printing, the pieces were washed using distilled water for 20 h and cured using a green lamp (36 W, 525 nm) for 2.5 h. In addition to the selected compositions, the incorporation of β-TCP and nHA was explored. In particular, we incorporated β-TCP and nHA in a proportion of 30 wt% in the unpolymerized resin (2:1% *w*/*w*, nHA:β-TCP). These bioceramics were added to the resin mixture to increase its bioactivity and improve its mechanical performance. The mixing was performed using an ultrasonic processor to achieve a homogenous mixture.

#### 2.3.3. Synthesis of β-TCP

The synthesis of β-TCP was carried out using the wet precipitation method reported by Sani et al. [37] who used a solution of calcium chloride dihydrate CaCl_2_·2H_2_O (0.3 M) and sodium phosphate dodecahydrate Na_2_HPO_4_·12H_2_O (0.3 M), both prepared in deionized water. The Na_2_HPO_4_·12H_2_O dissolution was incorporated in a beaker and slowly added dropwise to CaCl_2_·2H_2_O under constant magnetic stirring. The solution was left stirring for an hour at room temperature. Then, a 25% ammonium hydroxide solution was added drop-by-drop until it reached a pH of ~8 and was left without agitation for 24 h. The obtained residue was collected and washed thrice with chromatography water and dried at 70 °C for 21 h. Finally, the sample was annealed at 1000 °C for 5 h. The product was then characterized using ATR-FTIR, FE-SEM, and XRD.

#### 2.3.4. Evaluation of In Vitro Cytocompatibility

MC3T3-E1 cells (ATCC, UK) were cultured in DMEM, containing 10% FBS, 10 mM of HEPES, 100 U/mL penicillin, 100 µg/mL streptomycin, and 2.5 µg/mL amphotericin B. The 5 × 10^4^ cells were seeded directly into a piece of polymeric printed disk of 5 mm × 5 mm × 3 mm size placed in a single well of a 48-well cell culture plate. The medium was replaced every three days, and the cell viability was determined at 1, 3, and 7 days of incubation by using the AlamarBlue HS^®^, a Resazurin-based test, as an indicator of cellular health. After 2 h of incubation with the AlamarBlue HS reagent at 37 °C in a humidified air atmosphere containing 5% CO_2_, the medium was collected from the samples, and the absorbance was measured at 570 nm using a microplate reader. MC3T3-E1 cells were seeded onto samples with coverslips (12 mm diameter) at 1 and 3 days. Cells were rinsed in phosphate-buffered saline (PBS; 10 mM Tris (pH 7.4), 100 mM NaCl, 5 mM KCl), and cells were fixed in 4% paraformaldehyde solution for 15 min at room temperature, washed with PBS three times, and permeabilized with 0.5% Triton X100 for 10 min at room temperature and labeled with Phalloidin (Biotium, Fremont, CA, USA, CF^®^594) for 20 min, with the cells mounted; the slides were air-dried in the dark and mounted in Duolink In Situ Mounting Medium with DAPI (Sigma, DUO82040) for 15 min. The cells were visualized using a Zeiss–Colibri epifluorescence microscope with a 40× objective. The images were analyzed using the Fiji software (Image J, version 1.53g, 2020).

#### 2.3.5. Alkaline Phosphatase (ALP) Assay

To measure the ALP enzyme, the 1 × 10^5^ of MC3T3-E1 cells were seeded directly into a piece of polymeric printed disks of 5 mm × 5 mm × 3 mm size placed in a single well of a 24-well cell culture plate in triplicate. The medium was replaced every three days, and the ALP was determined at 7 and 14 days of incubation using a kit (ABCAM, ab83369), following the product’s manual. Briefly, the samples and pNPP substrate (p-nitrophenyl phosphate) were mixed and reacted for 60 min, and the reaction was ceased with a stop solution. The microplate was then read at 405 nm wavelength.

##### Statistical Analysis

The cytocompatibility and ALP assay statistical analysis was performed using the Origin Pro 2021 software, by one-way ANOVA, and with comparisons between samples and control using Tukey’s analysis with significance *p* < 0.05, *p* < 0.01, and *p* < 0.001.

#### 2.3.6. Chemical Characterization

Chemical characterization was performed to determine the composition of the final 3D-printed parts as a function of the initial resin mixtures via FT-IR, Raman, and UV–visible spectroscopies using the liquid, un-polymerized resin and the cured cylinders (completely polymerized). The chemical structure of the samples was studied in an FT-IR Nicolet iS 5 in the attenuated total reflection (ATR) mode. A 5 µL drop of unpolymerized resin and a small pulverized cured cylinder were used. Two types of UV–visible measurements were performed on the fabricated resin diluted with DMSO in 1:4 ratio, i.e., the amount of light absorbed in the 200 to 800 nm wavelength range and the intensity variation for 10 min at 525 nm wavelength.

#### 2.3.7. Contact Angle and Superficial Tension Measurements

Contact angle measurements were performed using a Theta optical tensiometer from Attension-Biolin Scientific (Gothenburg, Sweden), by adding 4 μL of the designed resins and a commercial biocompatible resin to the printing platform. To obtain the surface free energy (SFE), contact angle measurements were performed over the solid samples using 4 μL of water, glycerol, and diiodomethane as liquid phases. Additionally, to establish the behavior of the fabricated resins during the printing process, surface tension was measured using the pendant drop method, in which water was used as the heavy phase and air as the light phase. 

#### 2.3.8. Mechanical Strain–Stress Tests

Mechanical tests were also carried out to obtain the strain–stress curves of the resins to assess their mechanical properties, such as stiffness (Young’s modulus) or the maximum fracture load. The cylindrical specimens were measured before the tests, of which the initial dimensions were characterized by length (L_0_) and radius (R_0_). The load (F) and displacement of the compression plates (Δ) were recorded during the test. The following equations can determine the axial stretch (λ): (5)$$\lambda =\frac{L_{0}+\Delta }{L_{0}}$$

Then, the engineering stress can be calculated as follows:(6)$$\sigma =\frac{F}{\pi R_{0}^{2}}$$

The material is subjected to a preload of 5 N for one minute and a speed of 2.36 mm/min (strain rate 0.1%) to bring the piece to its fracture point.

## 3. Results and Discussion

This study is centered on two different resin types based on the PET-RAFT polymerization methodology using green irradiation to accomplish compound curation. The polymers were designed to create scaffolds for cellular osteogenesis applications, particularly for bone graft implants. The first resin corresponds to a mixture between DMAEMA and AAm as monomers and DBTTC as the RAFT agent (PDAD-8), while the second reaction mixture includes HEMA and HPMA as main monomers and CDTPA as the RAFT agent (PHHC-3). In both cases, PEGDA_250_ was used as the crosslinking agent, TEOH was the reducing agent, EY was the photoinitiator, and RB was the photoabsorber. 

With the finality to increase the bioactive character and the mechanical response of the materials, two different bioceramics were utilized (nHA and β-TCP). These mixtures were used as resins for DLP printing to fabricate biocompatible and bioactive printed parts. Two types of pieces were printed, i.e., (1) flat circular disks as prototypes for printing parameter optimization and for carrying out biocompatibility tests, and (2) three-dimensional scaffolds with internal porous architectures by using the TPMS approach. Additionally, chemical and physical properties were studied before and after PET-RAFT polymerization to identify the chemical structural changes due to the polymerization process. Mechanical and biological studies were also carried out in both samples, with the finality of comparing these results with the data observed in bone tissue.

### 3.1. Rheological and Chemical Characterization of the Resin

Once the resins were synthesized, the first step was to characterize the as-synthetized resin to determine its printability. From a rheological point of view, although no formula or factor allows us to decide whether or not a resin is printable using DLP technology (as it is in the case of inkjet printing with the Ohnesorge number), some specific parameters and variables allow us to determine if the resin is within an appropriate range to be printed, like viscosity [38,39] and surface tension [40,41]. According to Becker et al., the dynamic viscosity must not surpass 5 × 10^3^ mPa⸱s; and lower viscosities allow printing because a new resin layer could be polymerized faster than the previous one [42].

In general, the Bond number (Bo) is used to compare the gravitational forces and the forces related to the surface tension of the fluid [43], and it could be helpful to determine if the liquid bridge formed in the interface of the metallic plate and the resin is stable. A high value of B_o_ indicates that the system is relatively unaffected by surface tension effects. In contrast, a low value (typically less than one) suggests that the surface tension of the fluid dominates the system, thus forming stable and axisymmetric liquid bridges. In our case, the values of the dynamical viscosity, surface tension, density, and Bond number are tabulated in Appendix A for each analyzed resin, resulting in low B_o_ values and indicating the formation of a stable liquid bridge.

On the other hand, it is essential to highlight that another critical factor that significantly affects the printing process of DLP technology is the adhesion force between the resin and the printing plate. An indirect way to characterize this adhesion is via the contact angle between the resin and the plate. Generally, a low contact angle is related to a high surface energy at the interface [43], indicating the presence of high adhesion forces between the resin and the metallic plate. In our case, the contact angles of both synthesized resins (Appendix A section) are low enough to indicate a suitable adhesion force between the resin and the metallic plate.

Once the rheological characterization and the validation of the resins for DLP 3D printing were accomplished, the chemical analysis of the compounds was carried out. Accordingly, FT-IR and Raman spectroscopies were carried out to detect the vibrational bands from the characteristic radical groups from the PHHC and PDAD resins. The relative intensities of the different vibrational bands analyzed from the resin’s components (monomers and crosslinking agent) are shown in Figure 1. The liquid resin (unpolymerized state) and the 3D-printed piece (polymerized state) were characterized in each case. These analyses were carried out to understand how the polymerization process occurs and how the reactive double bonds are consumed during the polymerization. For example, the band located at ~1730 cm^−1^ corresponds to the C=O stretching mode of the ester functional group [44], and, evidently, it is present with high intensity in all of the resins (both in the FT-IR and Raman spectra). No visible intensity or shape alteration is detectable in this peak before and after the polymerization, evidencing the integrity of these functional groups after the 3D-printing process.

In contrast to the C=O functional groups, the peak related to the C=C vinyl stretching mode located at ~1645 cm^−1^ [45] varies significantly during the polymerization process. According to Gonzalez et al. [46], Raman spectroscopy can monitor the monomer mixture’s polymerization reaction following the double-bond intensity variation from the methacrylate and acrylate functional groups (C=C stretching mode) present in the monomers. Thus, the vinyl bond conversion was determined from the normalized area under this vibrational band according to the following equation (Equation (7)): (7)$$C_{v}=\left(1-\frac{{\left(I_{C=C}\right)}_{p}}{{\left(I_{C=C}\right)}_{l}}\right)$$
where (I_C=C_)_p_ is the area under the vibrational band of the C=C group after polymerization (printed part), and (I_C=C_)_l_ is the band area of the liquid resin (unpolymerized). According to these results, the resin PHHC has a polymerization ratio of 73% and the PDAD resin of 69%. This analysis of the Raman spectra shows us that approximately 30% of the monomer does not polymerize during the printing process, which can be disadvantageous for the material biocompatibility since, in general, the presence of unreacted monomers can result in abrupt pH changes in the medium, therefore decreasing the cytocompatibility capacity of the material. To avoid this, an exhaustive washing was carried out (20 h) in addition to an extensive post-curing process (2.5 h) to eliminate all possible traces of unreacted monomer. To check that the piece was not cytotoxic, they were immersed in a culture media with phenol red; this reagent has the peculiarity of changing its color quickly if the medium becomes acid or basic (yellow or fuchsia, respectively). Our samples showed no color change in the culture media after 6 h of immersion.

In parallel, the bands located between 1475 and 1450 cm^−1^ and at ~1410 cm^−1^ are related to the scissoring and antisymmetric deformation of the CH_2_ and CH_3_ groups. Evidently, from the spectra, one of the bands tends to decrease when the resin polymerizes, while the band at ~1475 cm^−1^ tends to increase [46].

Another fundamental chemical characterization of the resin is related to the polymerization kinetics of the PET-RAFT process. UV–visible spectroscopy was carried out on the liquid resins (PHHC and PDAD) to measure the time required for their complete polymerization, i.e., to achieve a high monomer conversion. Firstly, a scan of the visible region was carried out to detect the optical behavior of the resin during the photopolymerization process. Figure 2 (left) illustrates that two 530 and 565 nm peaks are detectable. These peaks are assigned to the chromophore (from either the photoinitiator or the photoabsorber) responsible for its red color. The spectra also present a shoulder that starts at 490 nm, ascribed to the dimeric form of Eosin-Y [47]. To analyze the polymerization kinetics of the resin, the absorption of the same reaction mixture was measured continuously using a 525 nm light source for 10 min. As reported by Alvarez-Martin et al. [47], the main absorbance band decays under oxidative conditions, causing the initially red solution to lose its color, a natural process related to the degradation of the chromophore. According to our results, in both cases (for PHHC and PDAD resins), just 90 s is enough to reduce the absorbance band to less than 20% of its initial intensity, indicating that the PET-RAFT process is almost concluded (Figure 2, right).

### 3.2. Synthesis of the Bioceramic Additive Compounds nHA and β-TCP

nHA and β-TCP powders were added to the resin as bioceramic fillers to increase the scaffolds’ bioactivity and improve their mechanical properties. Firstly, the synthesis of nHA was carried out using the previously reported wet chemical processing method [48], based on a solution of mono-ammonium phosphate (NH_4_)H_2_PO_4_ (0.6 M) and calcium nitrate tetrahydrate (Ca(NO_3_)_2_·4H_2_O) (1 M), prepared in equal volumes of deionized water. The final product was characterized using FTIR, FE-SEM, and XRD [49]. These results were compared with the commercial nHA. The FTIR analysis confirmed the presence of the characteristic functional groups of nHA, similar to XRD, which shows concordance with the one reported by RRUFF (ID: R060180 or R130713).

In the case of β-TCP, the synthesis was carried out following the methodology reported by Sani et al. [37]. This method employs a solution of sodium phosphate dibasic dodecahydrate (Na_2_HPO_4_·12H_2_O) (0.3 M) and calcium chloride dihydrate (CaCl_2_·2H_2_O) (0.3 M). The powder obtained after the calcination process was chemically and physically characterized. The FT-IR spectra (Appendix A) show 557 cm^−1^ and 612 cm^−1^ bands corresponding to the O–P–O vibrational modes. Also, the peaks at 1000 cm^−1^ and 1026 cm^−1^ and the band between 1065 cm^−1^ and 1085 cm^−1^ are assigned to PO_4_^−3^ bending and stretching modes, characteristic of β-TCP structure [50]. On the other hand, the absence of the bands located between 1420 cm^−1^ and 1480 cm^−1^ (CO_3_^−2^ groups) evidenced the success of the annealing process; in general, higher annealing times should produce lower amounts of CO_3_^−2^, like in this case [51]. The XRD data and the Miller indices obtained from the analysis are also in good agreement with those reported both by JCPDS (ID: 09-0169) and RRUFF (ID: R120115), with hexagonal lattice miller indexes of (101), (006), (015), (110), (113), (202), (205), (1010), (125), (300), and (200). See Appendix A.

### 3.3. Preparation of 3D-Printed Parts Using Biocompatible Resins and Incorporating Bioceramics

Once the newly synthesized bioceramics are characterized, biocomposites are prepared, including these inorganic compounds within the designed resins. For this purpose, 20% *w*/*w* of nHA and 10% *w*/*w* of β-TCP (2:1 ratio) were added to each resin to maintain 30% *w*/*w* of solid particulate material. Previous studies demonstrated that above 30% *w*/*w* of particulate within the resin increases the viscosity and reduces the sedimentation times, thus avoiding the fabrication of well-defined parts [52]. More precisely, with 30% *w*/*w* of particulate material, a viscosity of 19.9 mPa.s was measured for the PDAD resin and 8.8 mPa.s for the PHHC resin, with a sedimentation time of around 5 min. A dispersing agent (DisperBYK) was added to the solution at a concentration of 0.5% *w*/*w* to increase the sedimentation time. DisperBYK is based on a solution of alkylammonium salt of a low-molecular-weight polycarboxylic acid polymer whose function is to act as a surfactant that reduces the surface tension of the solution when mixed with the particulate material. As a result, small micelles are formed, which are more stable over time, taking longer to decant, increasing from 5 min to almost 18 min using the dispersing agent.

Good quality and structurally stable pieces were obtained when printing these resins with the bioceramic compounds using up to 30% wt particle material, constituted by 20% wt nHA, β-TCP, and 10% wt salts. Furthermore, using FE-SEM micrographies of the printed parts, it was possible to demonstrate that the inorganic particles were homogeneously distributed both on the surface and in the internal zones of the printed piece. High-magnification FE-SEM images were taken to determine the size of the nHA and β-TCP particles, obtaining diameters of 58.13 ± 0.584 nm and 1.40 ± 0.343 μm, respectively.

Evidently, the diameters are considerably different, with β-TCP being several times larger than nHA (more than 24 times); see Figure 3c. This explains why, in Figure 3a,b, two types of different particles in the material surface, large crystals with defined edges and small particles distributed homogeneously, are observed in the material. To confirm this observation, EDS images were acquired in these areas (Figure 3d). The scanned surface shows large crystals and small particles with similar elemental compositions attributable to nHA and β-TCP structures (in this case, exemplified by the signal Kα_1_ of the phosphorus (P), colored in red). At the same time, the organic phase of the piece (polymeric resin) is distributed homogeneously throughout the entire background of the image (Kα_1,2_ signal of carbon (C), colored in blue). This implies that the nHA particles are homogeneously distributed and embedded (encapsulated) within the polymer matrix (resin). In contrast, the β-TCP particles are relatively concentrated, forming clusters and crystals of sizes below ~1.25 µm.

### 3.4. Mechanical Properties of the PHHC and PDAD 3D-Printed Resins

Provided the homogeneous mixture between the bioceramic particles and the polymeric resin, several pieces with different internal structures were printed to test the material’s mechanical properties. As mentioned before, the compression velocity was maintained slowly enough to measure the material under a quasistatic regime, implying that the mechanical parameters obtained for the material correspond to a non-dynamical representation. Accordingly, as depicted in Figure 4a for the PHHC resin, the static Young modulus for the pieces fabricated using the gyroid structure is slightly higher than the diamond structure (0.94 MPa for gyroid and 0.84 MPa for diamond, Table 3) for identical filling density. This behavior is expected, considering the previous studies of Feng et al. [24] and Kladovasilakis et al. [53]. Interestingly, when including the bioceramic particles (nHA and β-TCP), the mechanical resistance of the material does not seem highly affected despite the large amount of particulate matter added to the mixture. This could probably be related to the chemical structure of the PHHC resin, which could present a higher interaction with the bioceramic compounds due to polar characteristics. nHA is fundamentally hydrophilic due to OH groups on the surface (H bonds) [54], favoring the interaction with PHHC (more hydrophilic than PDAD, see Appendix A), thus presenting significant shear stress at the particle–polymer interface and decreasing the mechanical resistance. 

In the case of the PDAD compound (Figure 4b), the situation is different. The gyroid structure presents a higher mechanical performance than that of the diamond structure, similar to PHHC (1.66 MPa for gyroid and 1.31 MPa for diamond, Table 3). However, when nHA and β-TCP particles are included, the Young modulus decreases considerably (0.94 MPa). In general, stiffness or Young’s modulus can be enhanced by incorporating micro- or nano-particles due to the higher stiffness of rigid inorganic particles compared to polymer matrices. However, the strength is closely tied to the effective stress transfer between particles and the matrix (shear stress transferred). Well-bonded particles allow the applied stress to be efficiently transmitted from the matrix to the particles, improving strength. Conversely, adding poorly bonded micro-particles leads to strength reductions [55].

Another important conclusion that could be derived from the strain–stress curve analysis is when bioceramic particles are added to the material. However, Young’s modulus is not strongly altered (a slight decrease in the case of PDAD); however, the shape of the curve changes considerably. We notice that, in both cases, for PHHC and PDAD, the curve of the materials with the gyroid or diamond structure initially shows an elastic linear deformation and then enters a viscoelastic zone. But curves with particulate bioceramic material always remain in an elastic deformation zone, at least up to 20–25% strain. This implies that although the material does not considerably alter its elastic modulus, it does become more rigid due to the inclusion of the bioceramic particles.

To exemplify it more clearly, in Table 3, the values of the strain and stress at fracture were also shown. In both cases (PHHC and PDAD), the stress at the moment of fracture increases for samples with embedded particles. At the same time, the deformation remains more or less at the same value, demonstrating a greater degree of material rigidity [55].

### 3.5. Biocompatibility Evaluation of the Designed Resins

MC3T3-E1 cells were cultured for up to 7 days, and then, AlamarBlue analyses were carried out to determine the cellular viability by analyzing the metabolic activity of the cells. ANOVA one-way statistical analysis was carried out for each group (1, 3, and 7 days) using the null hypothesis that the mean of the distributions is equal. This analysis results in *p*-values, indicating no statistical significance between the samples, as shown in Figure 5. In all cases, the material presents an acceptable behavior regarding its biocompatibility close to those values obtained with the control sample. Comparing the AlamarBlue absorbance generated by the different samples against the control (wells without material), it is possible to conclude that the material maintains cellular viability when control wells are compared to samples until 7 days. Epifluorescence microscopy in Figure 6 shows that cells tend to adhere to the material’s surface (with incubation times of 1 and 3 days), being detectable by staining the cytoskeleton and nucleus, which is indicative of the high biocompatibility of the material. In this case, the cells are extended on the surface, which indicates good cell growth and a future increase in the density of the cell layer formed on the material. If the surface did not generate good adhesion with the cells, the cells would have a spherical appearance without a developed cytoskeleton.

### 3.6. Osteogenic Differentiation (ALP Assay)

The ALP activity obtained for the PHHC and PDAD resins, and therefore, their capacity for the osteogenic differentiation of MC3T3 cells, is shown in Figure 7. Cells cultured on PDAD resins at 14 days had significantly higher ALP activity than the control (*p* < 0.05) but lower than that of nHA. Several studies have demonstrated the influence on the osteoinductive capacity of nHA when incorporated into polymeric matrices [56,57] or in coatings [58].

## 4. Conclusions

This study aims to fabricate biocompatible scaffolds for bone replacement using a polymer-based DLP resin for 3D printing. The approach’s novelty relies on the resin polymerization method and the complex inner structures included in the final printed piece. Random radical polymerization has emerged as the predominant mechanism for the DLP process due to its ability to tolerate a wide range of monomers and functional groups. Nevertheless, this type of polymerization produces inert or “dead” polymers that cannot be further modified. An exciting approach to impart a “living character” is PET-RAFT polymerization. Two distinct types of resins were synthesized based on PEGDA_250_ as a crosslinking agent. Thus, the first resin is constituted by DMAEMA and AAm as monomers and DBTTC as the RAFT agent. HEMA and HPMA were used as base monomers for the second resin and CDTPA as the RAFT agent. Eosin Y was used as a photoinitiator, Rose Bengal (RB) was used as a photoabsorber, and TEOH was used as a reducing agent for both reaction mixtures. Then, nHA and β-TCP particles were synthesized and added to the resin to improve the material’s bioactivity and mechanical resistance of the final printed piece. In parallel, different TPMS structures (diamond and gyroid) were also included in the material’s internal structure to increase the scaffold’s porosity and pore interconnectivity without strongly affecting its mechanical properties and resistance.

The synthesized resin was characterized by its printability via its rheological behavior. Then, physico-chemical and mechanical tests were performed on the printed parts with the finality to ensure their performance as the possible materials for TE applications. Finally, biocompatibility and osteogenic differentiation assays were performed on the scaffolds. Using the rheological results, it was possible to determine which mixture compositions are the most suitable to be used as DLP resins by determining their contact angle and Bond number. Once the two resins were selected, FTIR and Raman spectroscopies were carried out to determine the chemical structure of the mixture and to detect the polymerization degree of the final printed piece. The results demonstrated that almost 70% of the resin polymerizes during the printing process and practically 100% polymerizes after washing and curing processes, showing no visible cytotoxicity when submerged in the culture media. The mechanical resistance of both resins was also determined with and without the inclusion of TPMS inner structures (diamond and gyroid) and with and without the inclusion of nHA and β-TCP bioceramics. The results show that G-TPMS presents better results regarding mechanical resistance increase. However, including bioceramic material in the mixture does not alter the considerable mechanical resistance, but it has an essential effect on the biocompatibility and bioactivity increase of the material.

The future insights or perspectives related to the development of this research are mainly focused on testing these materials on different tissues, such as cartilage. The study’s mechanical and biocompatibility results indicate that the material could be used to these ends. On the other hand, our next goal in applying this technology is to extend our tests to in vivo studies in murine models. We are also interested in increasing the interaction between the bioceramic particle and the polymeric resin via the inclusion of a silane-coupling molecule (APTMS, TSM, APTES, among others) and, therefore, increase the mechanical resistance of the printed pieces when bioceramic particles are included in the resin mixture.

## Figures and Tables

**Figure 1 polymers-16-00472-f001:**
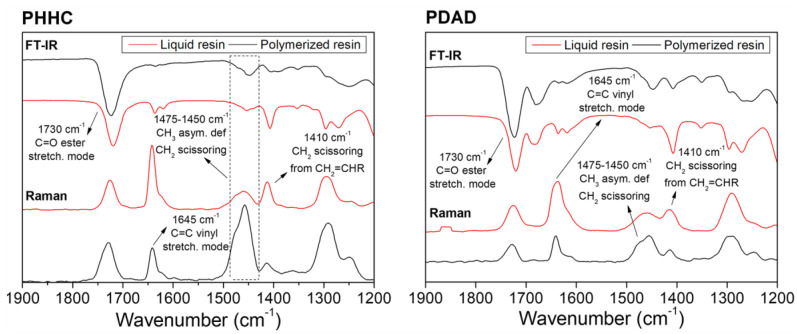
FT-IR and Raman spectra of the PHHC and PDAD resins before (liquid resin, red line) and after (polymerized resin, black line) the printing process.

**Figure 2 polymers-16-00472-f002:**
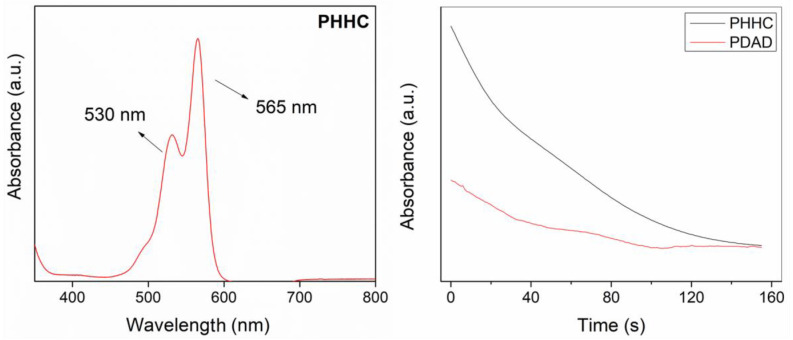
(**Left**) UV-vis spectra of the PHHC resin and (**Right**) absorption band reduction at 525 nm for the PHHC and PDAD resins over time.

**Figure 3 polymers-16-00472-f003:**
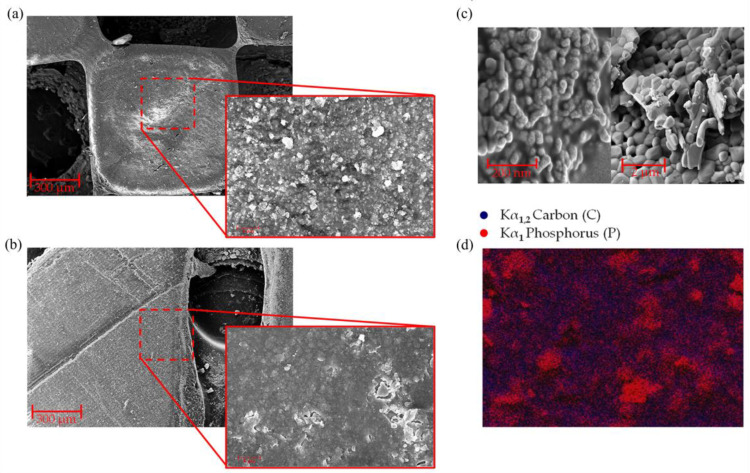
FE-SEM micrographs of the resin: (**a**) PHHC, (**b**) PDAD and EDX, (**c**) magnifications of FE-SEM of nHA (**left**) and β-TCP (**right**), and (**d**) EDS of PHHC.

**Figure 4 polymers-16-00472-f004:**
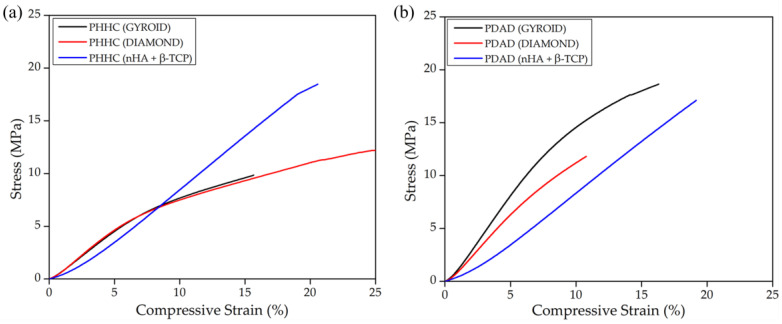
Stress–strain curves for the compressive mechanical tests on the cylindrical samples with gyroid TPMS structures fabricated with (**a**) PHHC and (**b**) PDAD resins.

**Figure 5 polymers-16-00472-f005:**
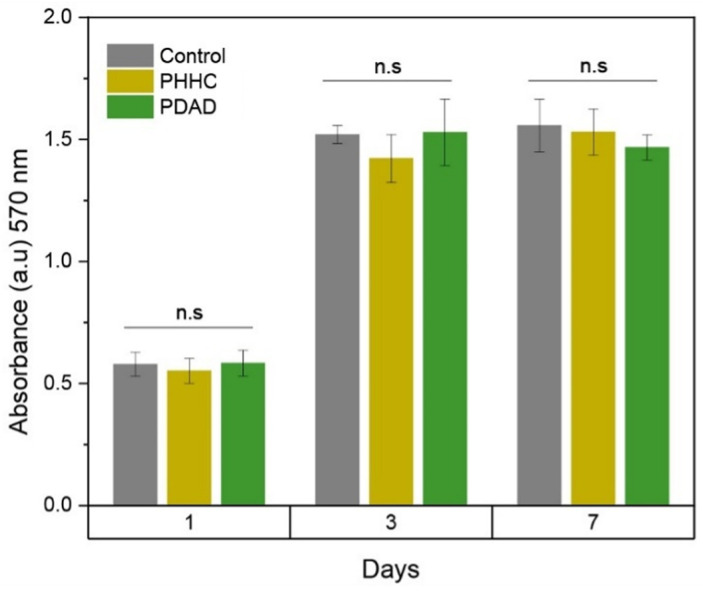
Cell viability studies obtained from AlamarBlue analysis for the control without material and samples PHHC and PDAD for 1-, 3-, and 7-day incubation (n.s is non-significative).

**Figure 6 polymers-16-00472-f006:**
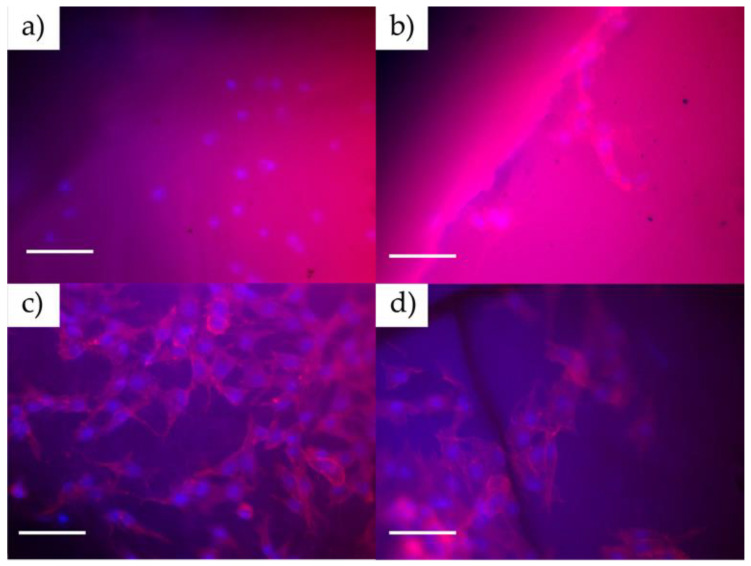
Epifluorescence micrographs of DAPI-stained and Phalloidin-labeled MC3T3 cells seeded on (**a**–**c**) PHHC and (**b**–**d**) PDAD at day 1, and at day 3. 40× magnification. Scale bar: 100 µm.

**Figure 7 polymers-16-00472-f007:**
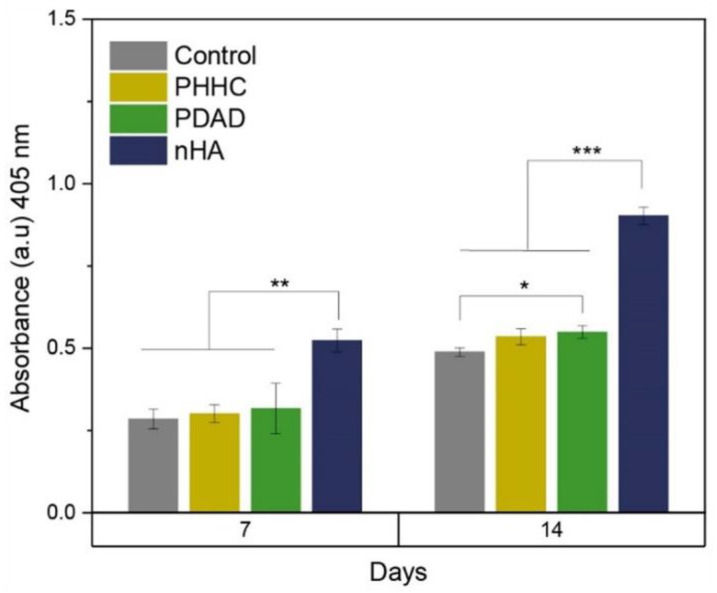
ALP activity in MC3T3 cell culture for PHHC and PDAD after 7- and 14-day incubation (* *p* < 0.05, ** *p* < 0.01, and *** *p* < 0.001).

**Table 1 polymers-16-00472-t001:** Amount of reactives used in the synthesis of PDAD.

N°	PDAD	PEGDA_250_ (mmol)	DMAEMA (mmol)	AAm (mmol)	DBTTC (mmol)	EY (mmol)	RB (mmol)
1	500:0:0	66.0	0	0	0.13	0.0013	0.0013
2	400:100:0	52.8	13.2	0	0.13	0.0013	0.0013
3	450:50:0	59.0	6.5	0	0.13	0.0013	0.0013
4	400:0:100	52.8	0	1.32	0.13	0.0013	0.0013
5	450:25:25	59.0	3.3	3.3	0.13	0.0013	0.0013
6	400:50:50	52.8	6.6	6.6	0.13	0.0013	0.0013
7	400:25:75	52.8	3.3	9.9	0.13	0.0013	0.0013
8	300:50:150	39.6	6.6	19.8	0.13	0.0013	0.0013
9	300:25:175	39.6	3.3	23.1	0.13	0.0013	0.0013

**Table 2 polymers-16-00472-t002:** Amount of reactives used in the synthesis of PHHC.

N°	PHHC	PEGDA_250_ (mmol)	HEMA (mmol)	HPMA (mmol)	CDTPA (mmol)	EY (mmol)	RB (mmol)
1	500:0:0	44	0	0	0.13	0.0013	0.0013
2	400:100:0	35.2	8.8	0	0.13	0.0013	0.0013
3	400:0:100	35.2	0	8.8	0.13	0.0013	0.0013
4	300:50:150	26.4	4.4	13.2	0.13	0.0013	0.0013
5	300:100:100	26.4	8.8	8.8	0.13	0.0013	0.0013

**Table 3 polymers-16-00472-t003:** Mechanical parameters (Young modulus, maximum strain, and stress at fracture) obtained from compression tests for PHHC and PDAD samples using gyroid TPMS structures.

	Resin	Young Modulus (MPa)	Strain @ Fracture (%)	Stress @ Fracture (MPa)
Gyroid	PHHC	0.94 ± 0.117	15.65 ± 5.40	12.7 ± 1.15
Diamond		0.84 ± 0.020	25.35 ± 0.976	11.3 ± 1.16
nHA + β-TCP		0.95 ± 0.032	23.33 ± 1.836	18.46 ± 1.24
Gyroid	PDAD	1.66 ± 0.24	16.31 ± 2.37	18.93 ± 0.005
Diamond		1.31 ± 0.04	10.21 ± 0.44	11.745 ± 0.32
nHA + β-TCP		0.94 ± 0.17	21.92 ± 2.07	19.78 ± 0.006

## Data Availability

Data are contained within the article and Appendix A.

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
