# Peer review of "Development of Biocompatible Digital Light Processing Resins for Additive Manufacturing Using Visible Light-Induced RAFT Polymerization"

_polymers, 2024, doi:10.3390/polym16040472_

Round 1

Reviewer 1 Report

Comments and Suggestions for Authors

This is an interesting paper well-organised by Mauricio A. Sarabia-Vallejos et al., I recommend the publication in this journal. Some minor comments: 

1) Some important key data can be involved in the abstract. 

2) Please add a scale bar in Figure 1 photos and figure 8. 

3) It will be better if the epifluorescence micrographs on different days can be provided, then the cell adherence tendency will be clearly displayed. 

Author Response

Response to reviewers:

Reviewer 1:

This is an interesting paper well-organized by Mauricio A. Sarabia-Vallejos et al., I recommend the publication in this journal. Some minor comments: 

  • Some important key data can be involved in the abstract. 

Response: Thanks for your commentary. We added key data in the abstract, as you suggested.

  • Please add a scale bar in Figure 1 photos and figure 8. 

Response: Thanks for your commentary. We added the scale bars, as you mentioned. It is important to highlight that Figure 1 is now on supplementary materials as Figura S1.

  • It will be better if the epifluorescence micrographs on different days can be provided, then the cell adherence tendency will be clearly displayed. 

Response: As you mentioned, we added some epifluorescence images in the mid-stage (1 and 3 incubation days). Thanks for your commentary.

Reviewer 2 Report

Comments and Suggestions for Authors The manuscript presented a comprehensive study of DLP printed, biocompatible polymeric materials and hybrid materials for potential application in bone repair. Highlighted is the use of RAFT polymerization for better control of the curing process. The manuscript is overall well prepared. A few points to be addressed: 1.  Line 85-87: The advantage of RAFT was addressed as non-deactivation of radical, and the polymer chain can further grow. Yet according to the work, the author uses DLP printing without further chain-growing, so judging from the results, the necessity of RAFT is not fully highlighted compared with the normal photoinitiators like TPO or something else. 

The authors can use a sentence to illustrate the necessity of the RAFT initiation system from the point of view of the cytotoxicity of conventional photoinitiators (e.g., the toxicity of TPO). Following this, the authors can state in another sentence that further modifications at the active end brought about by RAFT may be possible in the future.

2. Line 169: Better use light intensity mW/cm2 instead overall watt.

3. Figure 1b is somehow not mentioned in the text, at least say something since it's a big picture.

Author Response

Response to reviewers:

Reviewer 2:

The manuscript presented a comprehensive study of DLP printed, biocompatible polymeric materials and hybrid materials for potential application in bone repair. Highlighted is the use of RAFT polymerization for better control of the curing process. The manuscript is overall well prepared. A few points to be addressed: 

  1. Line 85-87: The advantage of RAFT was addressed as non-deactivation of radical, and the polymer chain can further grow. Yet according to the work, the author uses DLP printing without further chain-growing, so judging from the results, the necessity of RAFT is not fully highlighted compared with the normal photoinitiators like TPO or something else. 

The authors can use a sentence to illustrate the necessity of the RAFT initiation system from the point of view of the cytotoxicity of conventional photoinitiators (e.g., the toxicity of TPO). Following this, the authors can state in another sentence that further modifications at the active end brought about by RAFT may be possible in the future.

Response: As you mentioned, one advantage of the RAFT synthesis route is the high biocompatibility that presents most reactives. However, there is another crucial factor when selecting the RAFT method as the main synthesis route, and it is the fact that as being a reversible method, the bonds formed in the polymer are not as strong as those commonly obtained through traditional polymerizations, which implies that the materials formed via RAFT methods are easier to biodegrade, giving another functionality to the compound. As you mentioned, part of this information was added on pages 2 and 3; thanks for the commentary.

  1. Line 169: Better use light intensity mW/cm2 instead overall watt.

Response: Thanks for your commentary. The unit and the light intensity values were changed.

  1. Figure 1b is somehow not mentioned in the text, at least say something since it's a big picture

Response: Thanks for the commentary; indeed, there was no mention of Figure 1b in the text. Due to some commentaries from another referee, this figure is now in the supplementary material section as Figure S1.

Reviewer 3 Report

Comments and Suggestions for Authors

Summary:

The manuscript titled "Development of Biocompatible DLP Resins for Additive Manufacturing Using Visible Light-Induced RAFT Polymerization" focuses on creating advanced 3D printable resins for bone scaffolds. The authors employ poly(ethylene glycol diacrylate) (PEGDA) and various monomers, utilizing the PET-RAFT polymerization method. A key feature is the integration of triply periodic minimal surfaces (TPMS) to improve porosity and mechanical resistance without compromising the material's integrity. Additionally, bioactive particles enhance biocompatibility, and the inclusion of the photoabsorber Rose Bengal significantly boosts printing precision and resolution. The manuscript presents a detailed exploration of the materials' properties, including rheological behavior, chemical composition, and in vitro biocompatibility.

The manuscript is rich in experimental content and offers a detailed study on the topic. However, it suffers from poor logical organization, as evidenced by a lack of coherence between sections and excessive detail. Many details and experimental results are redundant or obvious, considering that many issues in DLP and SLA technologies have already been resolved or thoroughly researched. The article should focus more on directly citing relevant papers instead of accumulating superficial experimental results and discussing well-known content in the field. The reviewer suggests shortening the redundant parts and emphasizing the innovative aspects of the paper. Piling up experimental results can detract from the manuscript's value, making it hard for readers to quickly grasp the main points. Additionally, the language should be concise and focused, with strong logical coherence.

Detailed Comments:

1. Why did the authors feel the need to modify the Anycubic printer? The Anycubic Photon Mono4K is equipped with a 405nm UV lamp, which, with the correct photoinitiator, should suffice for curing. Why did the authors choose a more complex approach?

2. Based on the reviewer's experience, PEGDA is more soluble in alcohol and IPA and is widely used for cleaning purposes. Why did the authors opt to soak the prints in DI water for 20 hours post-printing?

3. The issue of printing accuracy discussed in Figure 1 is quite basic. The topic of photoabsorbers has been extensively discussed since the inception of SLA and DLP technologies. There's no need for extensive experimental discussion on this; it could be briefly mentioned.

4. The manuscript discusses too many issues, many of which, including viscosity testing, chemical characterization, and adhesion force, are fundamental. The logical flow of the article can be better.

Comments on the Quality of English Language

Can be improved by optimizing redundant content.

Author Response

Response to reviewers:

Reviewer 3:

The manuscript titled "Development of Biocompatible DLP Resins for Additive Manufacturing Using Visible Light-Induced RAFT Polymerization" focuses on creating advanced 3D printable resins for bone scaffolds. The authors employ poly(ethylene glycol diacrylate) (PEGDA) and various monomers, utilizing the PET-RAFT polymerization method. A key feature is the integration of triply periodic minimal surfaces (TPMS) to improve porosity and mechanical resistance without compromising the material's integrity. Additionally, bioactive particles enhance biocompatibility, and the inclusion of the photoabsorber Rose Bengal significantly boosts printing precision and resolution. The manuscript presents a detailed exploration of the materials' properties, including rheological behavior, chemical composition, and in vitro biocompatibility.

The manuscript is rich in experimental content and offers a detailed study on the topic. However, it suffers from poor logical organization, as evidenced by a lack of coherence between sections and excessive detail. Many details and experimental results are redundant or obvious, considering that many issues in DLP and SLA technologies have already been resolved or thoroughly researched. The article should focus more on directly citing relevant papers instead of accumulating superficial experimental results and discussing well-known content in the field. The reviewer suggests shortening the redundant parts and emphasizing the innovative aspects of the paper. Piling up experimental results can detract from the manuscript's value, making it hard for readers to quickly grasp the main points. Additionally, the language should be concise and focused, with strong logical coherence.

Detailed Comments:

  1. Why did the authors feel the need to modify the Anycubic printer? The Anycubic Photon Mono4K is equipped with a 405nm UV lamp, which, with the correct photoinitiator, should suffice for curing. Why did the authors choose a more complex approach?

Response: The main reason is that most of the photoinitiators and reactives used for printing with UV light are cytotoxic. However, in the range of the green light, several photoinitiators generate printed pieces with high biocompatibility. Besides, the methodology PET-RAFT was used as a synthetic route because, due to its reversible nature, it generates weak polymeric bonds compared to traditional radical photopolymerization, thus producing pieces with more potential as biodegradable materials. That is the reason for using a complex approach. Part of this information was added in the introduction section (pages 2 and 3).

  1. Based on the reviewer's experience, PEGDA is more soluble in alcohol and IPA and is widely used for cleaning purposes. Why did the authors opt to soak the prints in DI water for 20 hours post-printing?

Response: You are entirely correct; indeed, our first choice was to use isopropyl alcohol to remove all the possible traces of unreacted compounds, but the pilot tests show that the samples washed with alcohol present lower biocompatibility than those washed with distilled water. These results were not shown initially, but our optimization route derived from washing the pieces with distilled water for at least 20 hrs.

  1. The issue of printing accuracy discussed in Figure 1 is quite basic. The topic of photoabsorbers has been extensively discussed since the inception of SLA and DLP technologies. There's no need for extensive experimental discussion on this; it could be briefly mentioned.

Response: As you mentioned, this part was removed from the manuscript and added to the supplementary material. Thanks for your commentary.

  1. The manuscript discusses too many issues, many of which, including viscosity testing, chemical characterization, and adhesion force, are fundamental. The logical flow of the article can be better.

Response: Similar to what was answered in the previous comment, and as suggested, some of these mentioned sections were removed from the manuscript and added to the supplementary material. We consider the chemical characterization of the resin important, even more so considering the scope of Polymers journal, so we decided to keep this section in the manuscript. We hope that the response will fulfill your expectations.

Round 2

Reviewer 3 Report

Comments and Suggestions for Authors

The author has addressed most of the questions. The quality of the paper is improved.